# Textural Features of Mouse Glioma Models Measured by Dynamic Contrast-Enhanced MR Images with 3D Isotropic Resolution

**Karl Kiser \*, Jin Zhang and Sungheon Gene Kim**

Department of Radiology, Weill Cornell Medical College, New York, NY 10065, USA,
\* Correspondence: karl.j.kiser@gmail.com

**Abstract:** This paper investigates the effect of anisotropic resolution on the image textural features of pharmacokinetic (PK) parameters of a murine glioma model using dynamic contrast-enhanced (DCE) MR images acquired with an isotropic resolution at 7T with pre-contrast T1 mapping. The PK parameter maps of whole tumors at isotropic resolution were generated using the two-compartment exchange model combined with the three-site-two-exchange model. The textural features of these isotropic images were compared with those of simulated, thick-slice, anisotropic images to assess the influence of anisotropic voxel resolution on the textural features of tumors. The isotropic images and parameter maps captured distributions of high pixel intensity that were absent in the corresponding anisotropic images with thick slices. A significant difference was observed in 33% of the histogram and textural features extracted from anisotropic images and parameter maps, compared to those extracted from corresponding isotropic images. Anisotropic images in different orthogonal orientations demonstrated 42.1% of the histogram and textural features to be significantly different from those of isotropic images. This study demonstrates that the anisotropy of voxel resolution needs to be carefully considered when comparing the textual features of tumor PK parameters and contrast-enhanced images.

**Keywords:** radiomics; image texture; dynamic contrast-enhanced MRI; isotropic resolution





## 1. Introduction

A growing trend in medical imaging is the shift from qualitative to quantitative image assessment. Texture analysis has long been used as a tool for image analysis and classification through quantifying the spatial distribution of intensities in an image [1]. In medical imaging, texture analysis has gained popularity owing to its ability to describe tissue heterogeneity. Characterizing the complex microenvironment of solid tumors has been an obvious application for texture analysis, as increased tumor heterogeneity may be related to malignancy as well as resistance to therapies due to diverse and dynamic molecular subtypes [2–4]. With vast improvements in pattern analysis tools in recent years, texture features have been leveraged to build predictive models for treatment response from medical images in the field of radiomics [5–7].

While the potential of texture analysis has been well demonstrated, its adoption in clinical settings remains elusive, partly due to a lack of standardization that results in poor reproducibility as choices made at every step of the analysis affect the extracted feature values and predictive model performance [7–14]. In addition, variations in imaging protocols at the acquisition and reconstruction levels can also create non-biologically related differences in feature values [7]. Image spatial resolution is often cited as a significant source of variability in feature extraction. For instance, images with an anisotropic voxel resolution and thick slices, often observed in MRI, underperformed compared to those with thinner slices [8]. For texture analysis, medical images acquired at a non-isotropic resolution are typically resampled to an isotropic resolution in order to calculate 3D texture features, which can still obfuscate the potential predictive capabilities of the textual features [9]. Resampling

images at an isotropic voxel resolution was able to reduce the feature dependency on the spatial resolution in some studies [10,11], but not in others [12–14]. However, efforts to standardize acquisition methods and reporting have improved reproducibility between studies with anisotropic image acquisition methods [6,9].

Among various imaging modalities, dynamic contrast-enhanced (DCE) MRI and the estimated pharmacokinetic (PK) parameter maps have been widely used in radiomics analysis for cancer grading, tumor molecular subtyping, and predicting response to treatment [15–17]. DCE-MRI is one of the most commonly used imaging methods for cancer, and it provides rich information about the heterogeneous tumor microenvironment, which makes it a good candidate for texture analysis applications. On the other hand, DCE-MRI is also one of the MRI methods that could be quite susceptible to the variability of textual features related to the image spatial resolution because DCE-MR images are typically acquired with an anisotropic resolution with a slice thickness substantially larger than the in-plane voxel size to achieve an appropriate temporal resolution. However, it has not been well understood which textual features of DCE-MRI could be more affected by the anisotropy of the voxel dimension.

In this study, we investigated how non-isotropic voxel resolution affects texture features commonly used for DCE-MRI by comparing them with DCE-MR images acquired with an isotropic voxel resolution, using a mouse model of a brain tumor. The objective of this study is to identify radiomic features that are most sensitive to having a 3D isotropic resolution as opposed to a non-isotropic resolution in contrast-enhanced images and PK parameter maps. We also assessed how the textual features found significant in previous DCE-MRI studies differed between the images with isotropic and anisotropic resolutions.

## 2. Materials and Methods

### 2.1. Animal Model

Six to eight-week-old C57BL6 mice ($n = 16$) were inoculated with $1 \times 10^5$ GL261 mouse glioma cells suspended in 4 μL of saline solution using a Hamilton syringe for stereotactic intracranial injection. The mice were scanned once between post-injection days 15 and 22, when tumors were observed. The mice were treated in strict accordance with the National Institutes of Health Guide for the Care and Use of Laboratory Animals, and this study was approved by the Institutional Animal Care and Use Committee.

### 2.2. Data Acquisition

MRI scans were performed on a Bruker 7T micro-MRI system, consisting of a Biospec Avance III-HD console (Bruker Biospin MRI, Ettlingen, Germany) with an actively shielded gradient coil (Bruker, BGA-12; gradient strength, 600 mT/m) and a $^1$H four-channel phased array receive-only MRI CryoProbe with a volume transmit coil (Bruker, Ettlingen, Germany). During MRI scans, general anesthesia was induced by 1.5% isoflurane in the air. The animal's body temperature was maintained at $34 \pm 2$ °C during the scan.

The 3D ultra-short echo-time (UTE) sequence with the 3D Golden angle Radial Sparse Parallel (GRASP) MRI method was used for DCE-MRI scans to achieve an isotropic spatial resolution and to minimize the $T_2$* effect [18]. The scan parameters were TR/TE = 4/0.028 ms, image matrix = 128 × 128 × 128, and field of view = 20 × 20 × 20 mm$^3$. This sequence was continuously run to acquire 154,080 spokes (51,360 spokes per flip angle segment of 8°-25°-8°) for 10 min and 13 s. A bolus of Gadobutrol (Gadavist, Bayer) in saline at the standard dose of 0.1 mmol/kg was injected through a tail vein catheter after the first 60 s for the acquisition of pre-contrast images. Prior to the DCE scan, a 3D isotropic high-resolution $T_1$ map was obtained using the same 3D-UTE-GRASP sequence with variable flip angles (8°-2°-12°, 12,776 spokes for each flip angle, with total acquisition time = 153 s).

### 2.3. Image Reconstruction and PK Parameter Maps

A joint compressed sensing and parallel imaging reconstruction was implemented based on the 3D-UTE-GRASP algorithm [18]. The image reconstruction was conducted

with zero padding to achieve the reconstructed image matrix = $256 \times 256 \times 256$, providing a spatial resolution of about $0.078 \times 0.078 \times 0.078$ mm$^3$ and a temporal resolution of T = 5 s/frame (Figure 1a). As described in the above section on data acquisition, the middle part of the dynamic data acquisition was conducted with a different flip angle (25°), as opposed to 8° for the first and last parts, such that there were sudden signal level changes (dotted lines in Figure 1c). Arterial input function (AIF) was obtained using the Principal Component Analysis (PCA) method with estimation of the pre-contrast $T_1$ value during conversion of the two-flip angle DCE-MRI data to the Gd concentration curve as shown in Figure 1b [19]. The $T_1$ maps measured before DCE-MRI were used to manually segment the whole tumors, free hand, using Amira Software (Thermo Fisher Scientific, Waltham, MA, USA), from healthy tissue for 3D ROI selection, based on higher $T_1$ values in the tumor compared to the surrounding brain tissue. The pre-contrast $T_1$ maps were also used for the contrast kinetic model analysis that was carried out for all the voxels in the segmented tumor using the Two-Compartment-Exchange Model (TCM) [20] and Three-Site-Two-Exchange (3S2X) Model [19,21–23] (Figure 1c). Estimated from the model fit were five parameters, including interstitial space volume fraction ($v_e$), vascular space volume fraction ($v_p$), blood plasma flow ($F_p$), permeability surface area product ($PS$), and intracellular water lifetime ($\tau_i$), as shown in Figure 2. The volume transfer constant ($K^{trans}$) was calculated from $PS$ and $F_p$ ($K^{trans} = [1 - exp(-PS/F_p)] F_p$).

*2.4. Assessment of Isotropic versus Anisotropic Resolution Images*

In conventional DCE-MRI, an anisotropic voxel resolution is commonly used. This can be due to using a slice thickness larger than the in-plane voxel resolution as well as using partial sampling in phase encoding directions to reduce scan time. In this study, we assume the anisotropic resolution is only achieved by using a slice thickness larger than the in-plane voxel resolution, while the in-plane voxel resolution is isotropic without any partial Fourier sampling. We took averages of every thirteen slices (0.078 mm $\times$ 13 = 1.016 mm) in each orthogonal direction across the whole tumor ROI from the isotropic images acquired with 3D-UTE-GRASP to generate corresponding images with an anisotropic resolution ($0.078 \times 0.078 \times 1.016$ mm$^3$) (Figure 3). We also generated anisotropic images in the axial plane with resolutions of ($0.078 \times 0.078 \times 0.546$ mm$^3$) and ($0.078 \times 0.078 \times 0.234$ mm$^3$) in the same fashion. The above-mentioned contrast kinetic model analysis was carried out for the anisotropic dynamic images to generate the contrast kinetic parameter maps of the anisotropic images. There was no other image pre-processing used prior to the extraction of radiomic features.

Texture features were extracted using the open-source Python package PyRadiomics version 3.0.1 [24]. We included 93 features; First-Order Histogram Features ($n$ = 18 features), Gray Level Co-occurrence Matrix (GLCM; $n$ = 24), Gray Level Dependence Matrix (GLDM; $n$ = 14), Gray Level Run Length Matrix (GLRLM; $n$ = 16), Gray Level Size Zone Matrix (GLSZM; $n$ = 16), and Neighboring Gray Tone Difference Matrix (NGTDM; $n$ = 5). In addition to the contrast kinetic parameter maps, the last frame of the DCE-MR images was included as delayed contrast-enhanced images for extracting texture features. Following IBSI recommendations [9], all the anisotropic resolution images were interpolated to have an isotropic resolution of $0.078 \times 0.078 \times 0078$ mm$^3$ prior to extracting the 3D texture features. In this way, the input images to the PyRadiomics analysis had the same isotropic resolution. To isolate the effect of shape and volume on feature calculations, the ROI segmented from the isotropic image was transferred to the simulated anisotropic images for a given subject. Discretization of the dynamic ranges of the contrast-enhanced images and parameter maps was performed using a fixed bin width determined by the Freedman-Diaconis rule: $W = 2 (IQR) N^{-1/3}$, where $IQR$ is the interquartile range and $N$ is the number of pixels [25]. The default PyRadiomics configuration was used for all other texture feature extraction settings (see http://pyradiomics.readthedocs.io (accessed 16 May 2022) for further information).

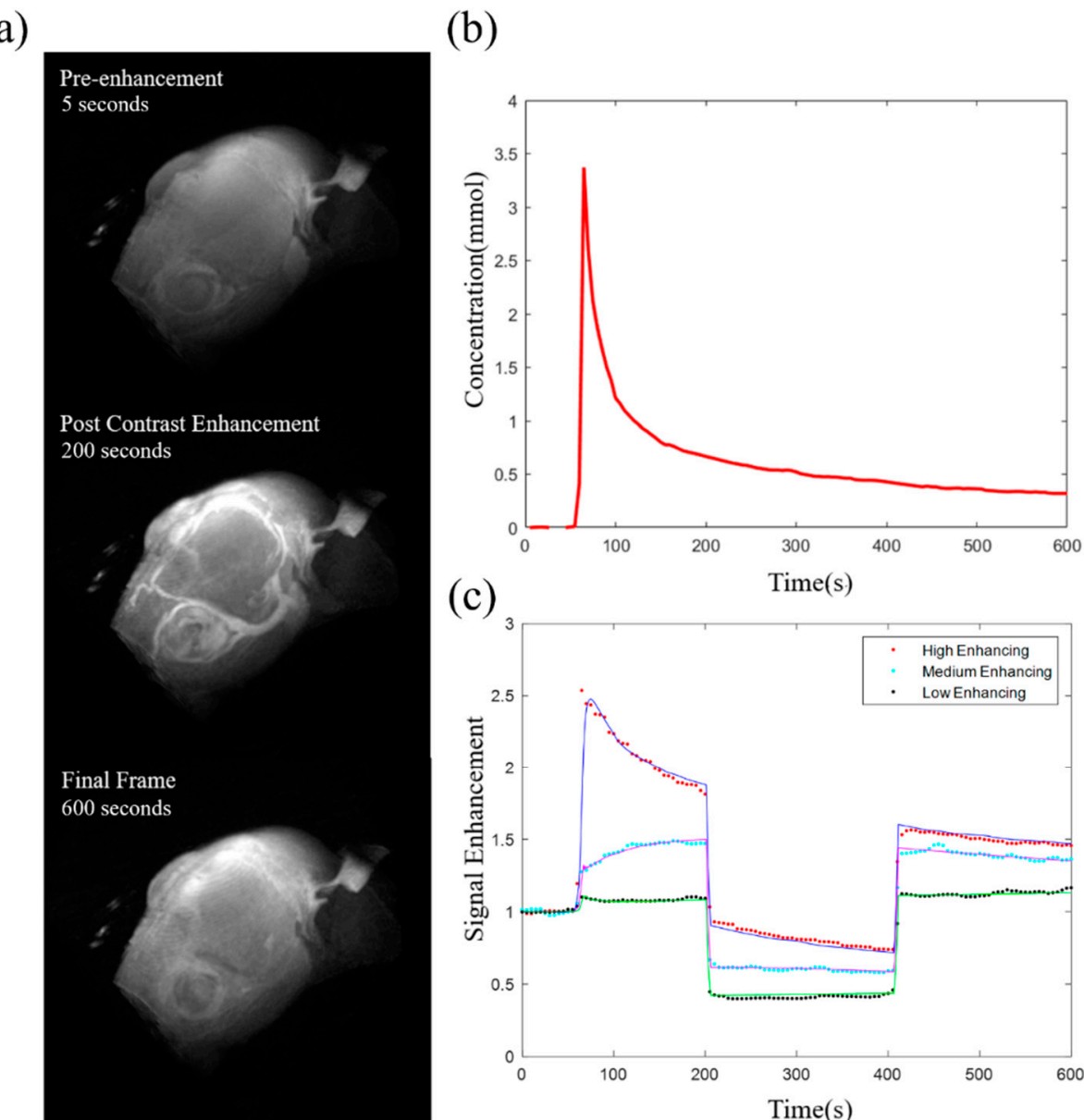

**Figure 1.** Example of acquired data from the 3D Dynamic Contrast-Enhanced (DCE) MRI. (**a**) Maximum intensity projections of whole head $T_1$ weighted DCE MR images for pre-contrast (5 s), early phase of post-contrast (200 s; 140 s post-contrast injection) and final frame (600 s). (**b**) Arterial input function derived from early enhancing vascular voxels selected with the Principal Component Analysis method. (**c**) Time intensity curves of three example tumor voxels with different enhancement levels (dotted lines), fitted with the two-compartment model and the three-site-two-exchange model (solid lines). The $K^{trans}$ values estimated from the model fit are 0.1043, 0.0559, and 0.0051 min$^{-1}$ for the high, medium, and low enhancing voxels, respectively.

Statistical analysis of texture feature differences between two images (isotropic vs. anisotropic images and different orientations of anisotropic images) was performed using the paired Student's *t*-test. To keep the family-wise error rate to 0.05 for each parameter, we used the significance level for a single hypothesis test of 0.00054 using the Bonferroni correction. We have also assessed the differences in the texture features of PK parameters that were found significant in previous DCE-MRI studies, as summarized in Table 1.

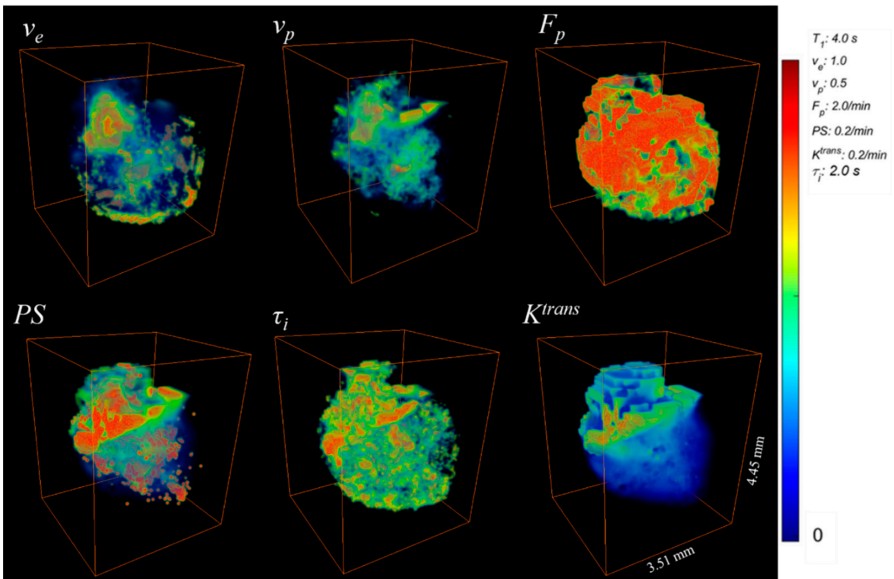

**Figure 2.** Volume renderings of the 3D contrast kinetic model parameter maps of the whole mouse glioma from Dynamic Contrast-Enhanced MRI data with an isotropic resolution. Included are the 3D maps for extravascular extracellular volume fraction ($v_e$), plasma volume fraction ($v_p$), blood flow ($F_p$), permeability surface area product ($PS$), transfer constant ($K^{trans}$), and intracellular water lifetime ($\tau_i$).

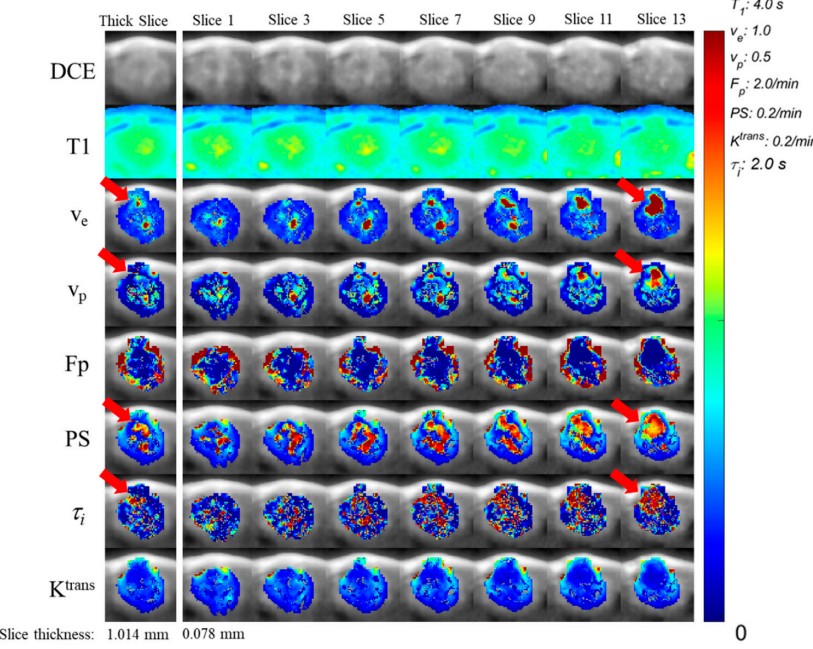

**Figure 3.** Axial isotropic parameter map slices and the corresponding anisotropic parameter map. The raw Dynamic Contrast-Enhanced MR images of the 13 slices (Slice 1–Slice 13, with a spatial resolution of $0.078 \times 0.078 \times 0.078$ mm$^3$) were vector averaged to simulate the anisotropic data for a thick slice, and then the contrast kinetic analysis was conducted to obtain the parameter maps shown on the left (spatial resolution of $0.078 \times 0.078 \times 1.016$ mm$^3$). Red arrows indicate large areas of high pixel values in the isotropic images but diminished values in the anisotropic images.

**Table 1.** Summary of pharmacokinetic parameters from DCE-MRI studies that were found significant by the radiomic analysis.

| Study | Acquisition Resolution | Cancer Type | Predicting | Parameter | Feature |
|---|---|---|---|---|---|
| W. Ma et al. (2018) [26] | $0.98 \times 0.49 \times 1.8$ mm | Breast Cancer | Ki-67 expression | Post-Contrast T1-w DCE | First-Order: Mean, SD, Skewness, and Kurtosis GLCM: Energy (Joint energy), Entropy (Joint entropy), Contrast, Correlation, Homogeneity (Inverse difference), and IDM |
| Y. Wang et al. (2019) [15] | $0.89 \times 0.89 \times 3$ mm | Prostate Cancer | Bone Metastases | Post-Contrast T1-w DCE | First-Order: 0.025 quartile GLCM: Auto correlation, Cluster prominence, Difference entropy, Dissimilarity, Homogeneity, IDM, and IDMNGLRLM: Short run low grey level emphasis and Short run high grey level emphasis |
| Thibault et al. (2016) [17] | $1 \times 1 \times 1.4$ mm | Breast Cancer | Response to Treatment | $K^{trans}$ $\tau_i$ | GLCM: Entropy difference, Contrast, Variance differences, and Inertia GLRLM: Gray-level nonuniformity and Long-run emphasis GLCM: Mean |
| | | | | $ve$ | GLCM: Contrast and Inertia |
| Xie T et al. (2017) [27] | $0.74 \times 0.53 \times 6.0$ mm | Glioma | Grading Ki-67 expression | $K^{trans}$ $ve, vp, K^{trans}, vp$ | GLCM: Energy (Joint Energy), Entropy (Joint Entropy), Inertia (Contrast), and Correlation IDM GLCM: Energy, Entropy, and IDM GLCM: Energy (Joint Energy) and IDM |
| Liu YYG et al. (2020) [28] | $0.6 \times 0.8 \times 3$ mm | Pituitary macroadenoma | Tumor 'Aggressiveness' via Heterogeneity in Vasculature | $K^{trans}$ $ve$ $K^{trans}, ve$ $Kep$ | First-Order: Skewness First-Order: Mean GLRLM: Long-run emphasis, Gray-level non-uniformity, High gray-level run emphasis, and Short run emphasis GLCM: Difference entropy GLRLM: Gray level non-uniformity and Run length non-uniformity |
| Zhou X et al. (2020) [29] | $1.4 \times 1.3 \times 4$ mm | Breast Cancer | Benign/malignancy | $K^{trans}, ve$ $Kep, ve$ $ve$ | GLCM: Entropy (Joint entropy) GLRLM: Short run low grey level emphasis GLCM: Cluster shade |
| | | | | $vp$ | GLCM: IDM |
| | | | Molecular Subtype | $K^{trans}, ve$ $K^{trans}$ $ve$ $vp$ | GLCM: Entropy (Joint entropy) GLRLM: Grey level non uniformity and Long run emphasis GLRLM: Short run emphasis First-Order: Entropy GLSZM: Zone percentage GLRLM: Short run high grey level emphasis and Short run low grey level emphasis GLSZM: High grey level emphasis |

## 3. Results

Figure 3 shows 7 slices out of the 13 consecutive slices (0.078 mm thick) in 3D isotropic high-resolution images that were averaged together to generate one image with a slice thickness of about 1 mm while maintaining the same in-plane resolution (0.078 mm $\times$ 0.078 mm). The PK maps of the thick slice case were generated after averaging the images to generate the anisotropic resolution images. This example illustrates possible discrepancies between the 3D isotropic image and the corresponding anisotropic resolution maps of the same tumor due to the partial volume effect in the anisotropic resolution images. There is a complex heterogeneity of the tumor morphology and parameter distribution captured in the 13 isotropic slices, which is evidently lost in the thick slice. Particularly of note are larger areas of high enhancement present in the isotropic images that are diminished or lost in the anisotropic image (indicated by red arrows). The distribution of voxel intensities may vary depending on the orientation of the tumor in anisotropic images relative to the high resolution in plane resolution. Furthermore, up-sampling anisotropic images to an isotropic resolution for 3D texture feature calculations has severe limitations in terms of restoring the spatial heterogeneity along the slice direction, as shown in Figure 4. This blurring of spatial information along the slice direction may have an impact on the texture features calculated using a 3D patch of voxels.

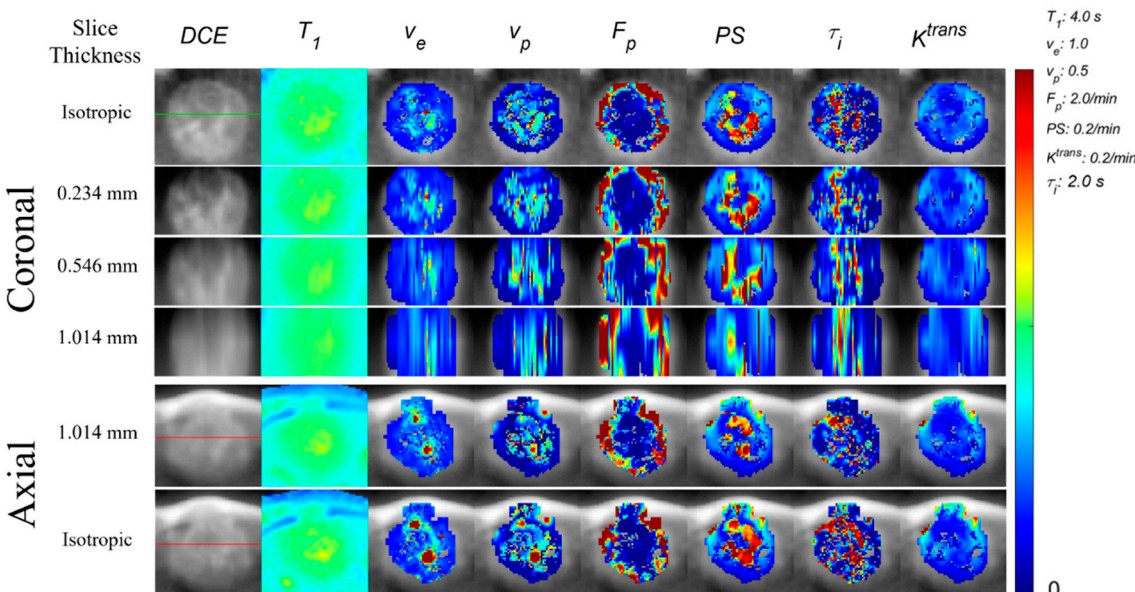

**Figure 4.** Impact of up-sampling anisotropic images along slice direction. The images are shown after up-sampling axial anisotropic images to isotropic resolution for the radiomics analysis. Spatial blurring of the image and parameters along the coronal direction is clearly shown. In contrast, such blurring is less remarkable in axial images (in-plane spatial resolution of $0.078 \times 0.078$ mm$^2$). The green line indicates the axial slice location, and the red line indicates the coronal slice location.

### 3.1. Isotropic vs. Anisotropic Resolution Images

The radiomic features were compared in terms of their percentage differences to minimize the inherently large differences in their magnitudes. The differences between isotropic ($0.078 \times 0.078 \times 0.078$ mm$^3$) and anisotropic ($0.078 \times 0.078 \times 1.014$ mm$^3$) images of all animal tumors are illustrated as heatmaps (Figure 5). Of the 75 texture features calculated for all parameter maps and DCE images, 31% demonstrated a significant difference, and of the 18 first-order histogram features for all images, 40% demonstrated a significant difference. Decreasing the slice thickness of the anisotropic images to 0.546 mm or 0.234 mm demonstrated similar patterns of percent differences but decreased the magnitude of the difference (Supplementary Figures S1 and S2). The percentage of significantly different texture features remained at 32% for images with 0.546 mm and decreased to 18% for 0.234 mm slice thickness, with significant histogram features reducing to 37% and 25%, respectively.

The distribution of significantly different texture features is not consistent among different images, with post-contrast T1-w images, $F_P$ and $\tau_i$ maps demonstrating much higher sensitivities to change in resolution (30.7–46.7% significantly different features at 0.234 mm) compared to $v_e$, $v_p$, $PS$, and $K^{trans}$ maps (0–5.3%) (Table 2). Similarly, histogram features from different parameter maps demonstrate different sensitivities to resolution change, where $v_e$, $v_p$, and $F_P$ maps with slice thickness of 0.234 mm show 0–16.7% significantly different features compared to isotropic resolution, and where $F_P$, $PS$ $\tau_i$, $K^{trans}$, and T1-w images range from 27.8–38.9%.

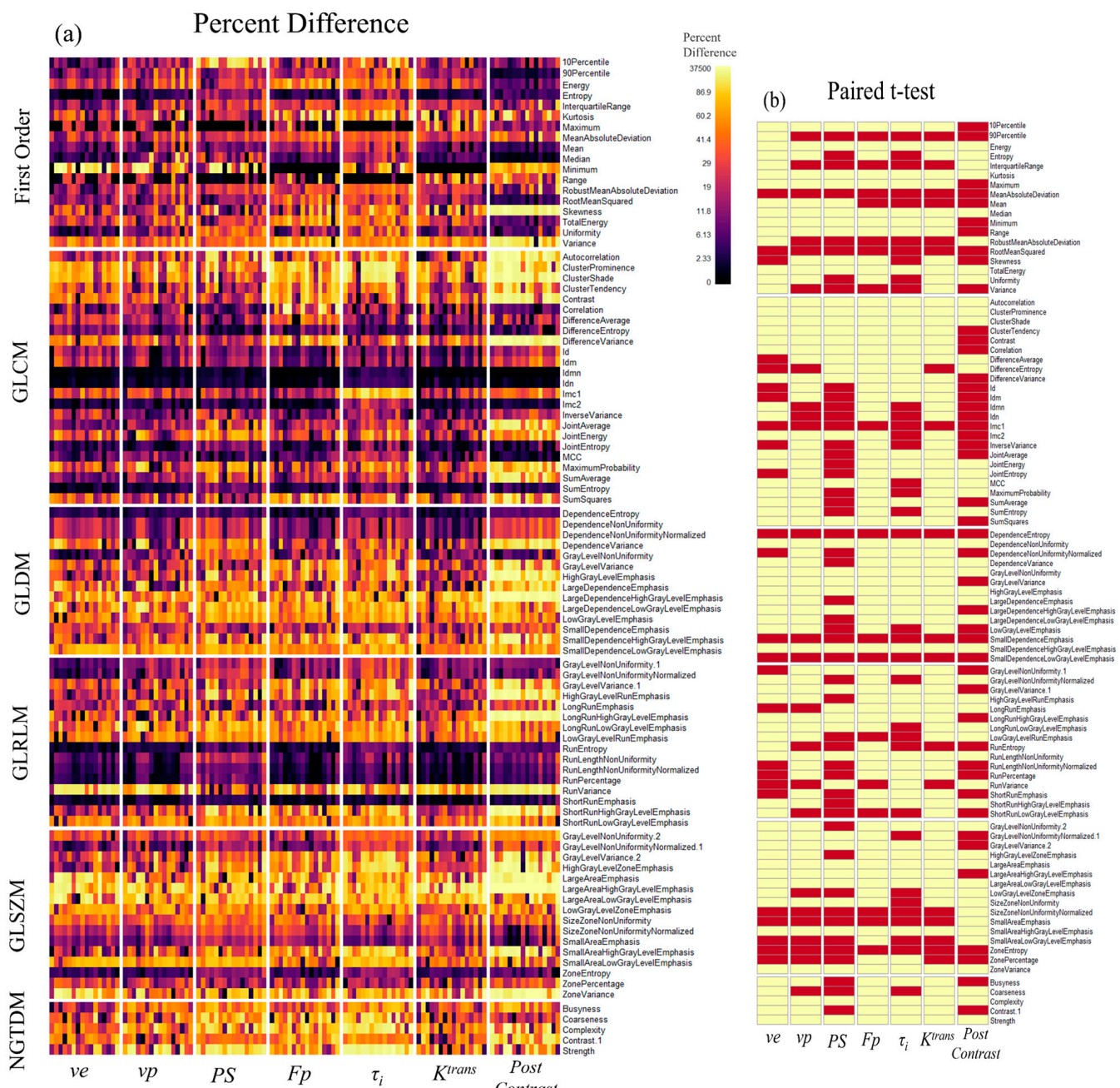

**Figure 5.** Percent difference and significance between features from isotropic and anisotropic (slice thickness of 1.014 mm) images. (**a**) A heatmap depicting the percent difference in histogram and texture features between isotropic parameter maps and up-sampled anisotropic maps (slice thickness of 1.014 mm in axial direction). (**b**) A heatmap showing the features that have a significant difference in radiomic features ($p < 0.00054$, depicted in red) between isotropic and anisotropic image resolution, which are over 32% of all features.

**Table 2.** The percentage of significantly different texture and histogram features among different parameters for images of various anisotropic resolutions and orthogonal slice directions compared to isotropic resolution. Post-contrast T1-weighted images, *FP*, and $\tau_i$ maps demonstrate higher sensitivity to changes in slice thickness (30.7–46.7% of significantly different features at 0.234 mm) compared to $v_e$, $v_p$, *PS*, and $K^{trans}$ maps (0–5.3%). Similarly, histogram features from different parameter maps demonstrate different sensitivity to resolution change, where $v_e$, $v_p$, and *FP* maps with slice thickness of 0.234 mm show 0–16.7% significantly different features compared to isotropic resolution, and where *FP*, *PS*, $\tau_i$, $K^{trans}$, and T1-w images range from 27.8–38.9%. Slice direction generally impacts the distribution of significantly different features similarly, with the notable exception of post-contrast T1-weighted images, where the sagittal direction impacts 58.7% of texture features compared to 20% and 10.7% of features for the axial and coronal directions, respectively.

| Percent of Features Demonstrating Significant Difference from Isotropic Resolution | | | | | |
|---|---|---|---|---|---|
| **Texture Features (*n* = 75)** | | | **Histogram Features (*n* = 18)** | | |
| 1.014 mm | 0.546 mm | 0.234 mm | 1.014 mm | 0.546 mm | 0.234 mm |
| $v_e$ | | | | | |
| 29.3 | 22.7 | 1.3 | 16.7 | 0.0 | 0.0 |
| $v_p$ | | | | | |
| 24.0 | 13.3 | 5.3 | 33.3 | 22.2 | 16.7 |
| FP | | | | | |
| 53.3 | 52.0 | 42.7 | 44.4 | 33.3 | 16.7 |
| PS | | | | | |
| 13.3 | 17.3 | 2.7 | 38.9 | 44.4 | 33.3 |
| $\tau_i$ | | | | | |
| 33.3 | 34.7 | 30.7 | 55.6 | 61.1 | 44.4 |
| $K^{trans}$ | | | | | |
| 16.0 | 17.3 | 0.0 | 33.3 | 27.8 | 27.8 |
| T1-w | | | | | |
| 48.0 | 66.7 | 46.7 | 55.6 | 72.2 | 38.9 |
| **Total** | | | | | |
| **31.0** | **32.0** | **18.5** | **39.7** | **37.3** | **25.4** |
| Ax | Cor | Sag | Ax | Cor | Sag |
| $v_e$ | | | | | |
| 29.3 | 28.0 | 28.0 | 16.7 | 33.3 | 11.1 |
| $v_p$ | | | | | |
| 24.0 | 18.7 | 24.0 | 33.3 | 38.9 | 38.9 |
| FP | | | | | |
| 53.3 | 60.0 | 57.3 | 44.4 | 50.0 | 38.9 |
| PS | | | | | |
| 13.3 | 18.7 | 26.7 | 38.9 | 38.9 | 50.0 |
| $\tau_i$ | | | | | |
| 33.3 | 32.0 | 33.3 | 55.6 | 72.2 | 66.7 |
| $K^{trans}$ | | | | | |
| 16.0 | 62.7 | 62.7 | 33.3 | 44.4 | 44.4 |
| T1-w | | | | | |
| 48.0 | 10.7 | 58.7 | 55.6 | 38.9 | 61.1 |
| **Total** | | | | | |
| **31.0** | **33.0** | **41.5** | **39.7** | **45.2** | **44.4** |

*3.2. Anisotropic Resolution Images in Different Orientations*

We also investigated how the radiomic features vary depending on the orientation of the anisotropic resolution, i.e., slice orientation. The heatmaps in Figure 6 show that the differences in GLCM features between anisotropic images in different imaging planes share similar patterns of differences and a similar range of significantly different features (32–34%). Furthermore, slice direction generally impacts the distribution of significantly different features similarly, with the notable exception of post-contrast T1-weighted images, where the sagittal direction impacts 58.7% of texture features compared to 20% and 10.7% of features for the axial and coronal directions, respectively (Table 2). Heatmaps for the complete feature set for anisotropic images in all 3 orthogonal imaging planes can be found in Figure 5 and Supplemental Figures S3 and S4.

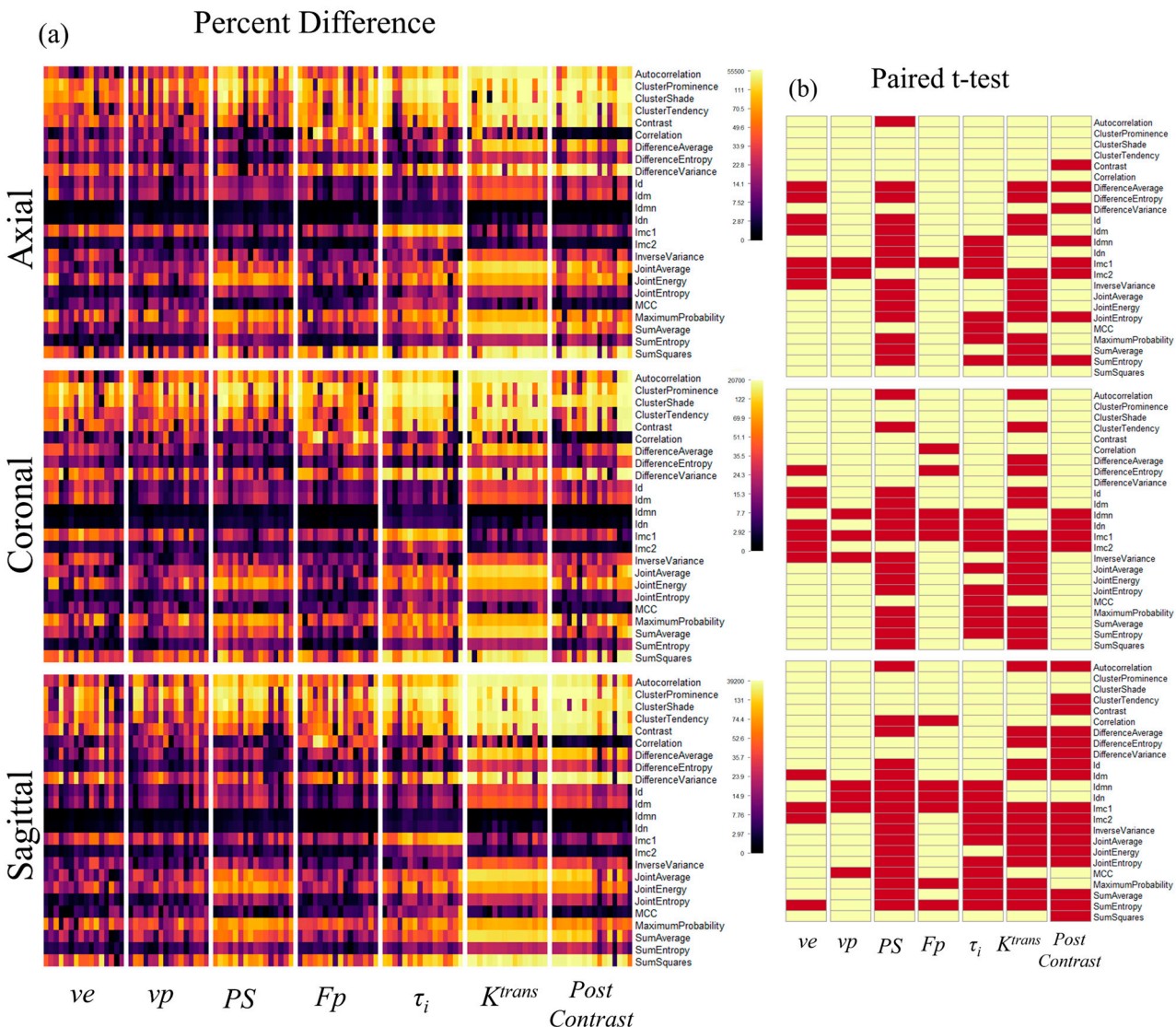

**Figure 6.** The percent difference and significance between Grey Level Cooccurrence Matrix (GLCM) texture features from isotropic and anisotropic images of different orthogonal orientations. (**a**) A heatmap depicting the percent difference in texture features calculated from GLCM between isotropic parameter maps and anisotropic maps of different orthogonal orientations. Anisotropic maps were simulated from the isotropic images in three orthogonal directions, all demonstrating similar patterns in the difference. (**b**) A heatmap showing the features that have a significant difference in radiomic features ($p < 0.00054$, depicted in red) between isotropic and anisotropic image resolution, which are between 32% and 34% of total features.

### 3.3. Texture Features Reported in DCE-MRI Studies

The radiomics features included in our present study have been used in other recent studies [15,17,26–29]. Table 1 presents a summary of the radiomic features found significant for predicting various aspects of tumors, such as malignancy and treatment response. Of the 40 features found to be predictive in the six different DCE MRI radiomic studies, the mean value of 17–25 of these features was found to be impacted by at least ±%10 when extracted from isotropic versus anisotropic images, and 17 features for all three slice thickness cases were found to be statistically significant, where all post-contrast T1-w features, two $K^{trans}$ features, and four $v_e$ features were impacted significantly.

## 4. Discussion

The principal aim of our study was to investigate the dependency of texture feature values on images acquired with an isotropic versus an anisotropic resolution. We found that over 32% of the texture features available in PyRadiomics were derived from PK parameter maps at isotropic resolution and were significantly different from those estimated from more commonly used images with an anisotropic resolution. Furthermore, of the 40 radiomic features found to be predictive in cancer imaging, we found 17 to be significantly impacted by anisotropic resolution (Figure 7). This suggests that it is critical to acquire images with an isotropic high resolution in order to use radiomic features as reliable biomarkers for cancer studies.

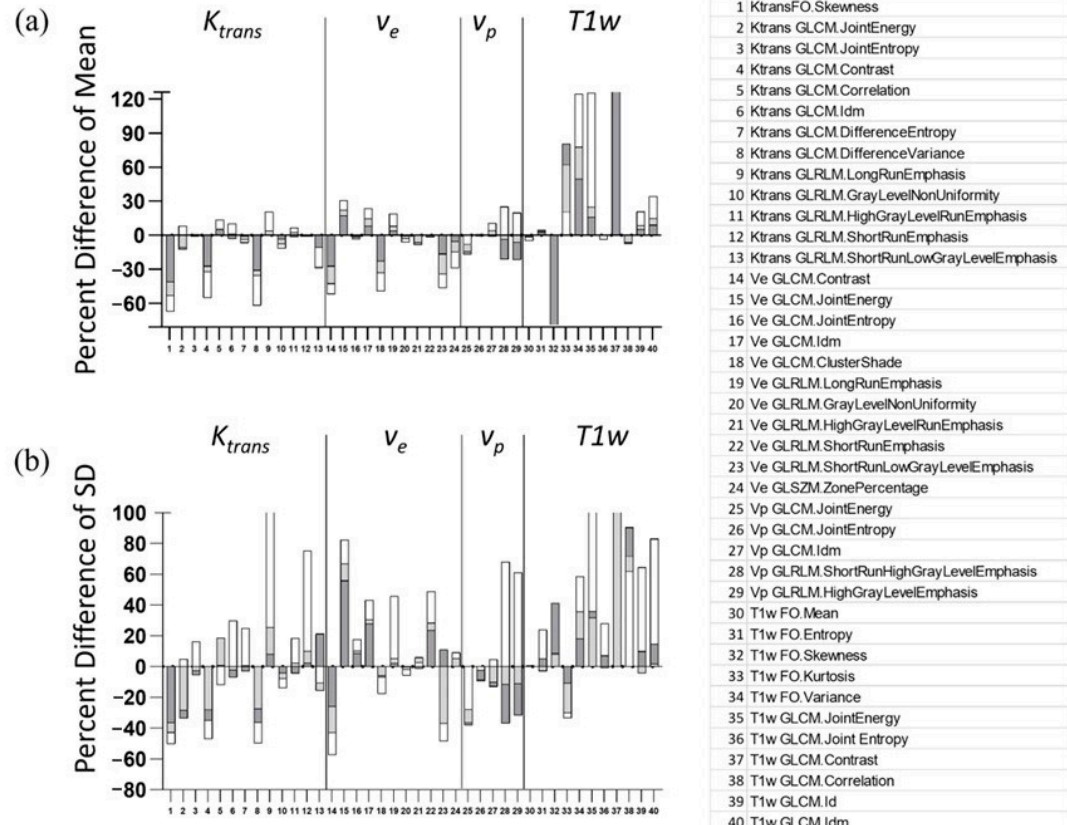

**Figure 7.** Bar plots representing the percent difference of the means (**a**) and standard deviations (**b**) of the 40 features used in recent studies listed in Table 1 show the difference between isotropic and up-sampled anisotropic images. The number of features with mean values differing by more than ±10% are 22, 24, and 32 for slice thicknesses of 0.234 mm (dark gray), 0.546 mm (light gray), and 1.014 mm (white), respectively. The number of features with SD values >10% is 22, 23, and 32, respectively. The percent difference of mean fell outside of the axis range for T1-w FO skewness (−217%, −268%, and −275%) and T1-w GLCM Contrast (165%,194%, and 253%). The percent difference of SD fell outside of the axis range for Ktrans GLRLM Long Run Emphasis (8%, 25%, and 196%), T1-w FO Variance (36%, 31%, and 375%) and T1-w GLCM Contrast (187%, 151%, and 232%).

A previous study [30] demonstrated that radiomic feature classification abilities are highly dependent on in-plane resolution, with improved classification from lower resolution images after interpolation to higher resolutions. The study suggests that adequate spatial resolution is needed for texture features to capture the relevant tissue microstructure. This is in line with the results of our study, which show that some features may have minor differences despite different slice thicknesses, as all images are resampled to isotropic resolution for 3D texture analysis.

A recent phantom study by Baeßler et al. [31] demonstrated poor reproducibility of close to half of the radiomic features calculated from T1-w images sampled at anisotropic resolutions. Similarly, our study found features from T1-w images to be most sensitive to resolution and thick slice orientation compared to PK parameter map images, despite the interpolation to an isotropic resolution. Baeßler argued that the reduced reproducibility of T1-w images versus FLAIR images from their study may be accounted for by a more subtle gray-level discretization from the spread of more gray levels across a smaller range in FLAIR images [31]. This may also be the case for our parameter map images, as their values are bound inherently to a physiologically relevant range. For this reason, using PK parameter maps or other quantitative imaging modalities may improve reproducibility in texture analysis and radiomic workflows.

Qualitatively, a high degree of detail is lost by the increased partial volume effect of images acquired at an anisotropic resolution with a thick slice compared to those acquired at an isotropic high resolution. Large areas of high intensity in isotropic images appear to be diminished or absent in the thick slice images as a result of the partial volume effect. This is most evident in Figure 3, comparing slice 13 of the isotropic parameter maps with the thick slice parameter maps for $v_e$, $v_p$, $PS$, and $\tau_i$. The smoothing and diminishing of high intensity areas likely account for the high variation in texture features measuring heterogeneity and intensity groupings. Regions of maximal abnormality, or 'hotspots', within the tumor parameter maps can be used for grading tumors [32], where the diminishing of these 'hotspots' in thick slice images likely impacts predictive capabilities and warrants further study. Additionally, several studies have found poor reliability in comparing thick slice image data with genomic and histopathology biomarkers, which represent many orders of magnitude difference in scale [33,34], making it challenging to validate image heterogeneity biomarkers against histopathology.

A non-trivial consideration for image texture analysis is the intensity discretization method, as there is currently no consensus on the best practice despite the high dependency of feature values on this choice [35,36]. For this study, we chose to discretize images based on the Freedman-Diaconis rule to statistically determine the bin-width size for each image between isotropic and anisotropic images [25,37]. Radiomic features have been shown to be highly dependent on tumor volume [38], which we hoped to reduce with a dynamic discretization approach such as the Freedman-Diaconis rule. Furthermore, discretization with bin-width selection has demonstrated reduced variabilities in texture feature values compared with choosing a bin number [39,40]. Additionally, selecting a bin-width instead of a bin number maintains the relative contrast of the image, which is especially important when pixel intensities are definite and tied to physiological parameters, such as in DCE-MRI PK parameter maps [24].

The lack of standardization of texture analysis in radiomic workflows, resulting in poor reproducibility, is a known issue, preventing its broader adoption in the clinic. The International Standardization Initiative (IBSI) [9] and Quantitative Imaging Network (QIN) [6] have weighed in, providing a standardization of feature definitions and guidelines on reporting radiomics studies to improve reproducibility. The choice of imaging platform and IBSI compliance have been shown to affect feature extraction and predictability, and by standardizing acquisition and post-processing settings such as resolution, variability in feature values is reduced [41]. IBSI guidelines are most impactful in retrospective studies where acquisition parameters are preestablished and reproducibility is dependent on reporting image processing settings. Moving forward, it is vital to standardize acquisition techniques to improve the reproducibility and power of quantitative imaging techniques like DCE-MRI through multi-institution initiatives where image analysis must be a focal point.

This study has a number of limitations. This study featured a limited number of subjects with a single tumor type. In future work, we plan to expand this study with different types of tumors. We are especially keen to investigate the impact of acquired image resolution on the predictive capabilities of texture features for response to treatment, which was not possible with the current cohort. All tumors in this study were manually

segmented, which may result in intraobserver variability, particularly in cases of anisotropic resolution [42]. However, the accuracy of lesion segmentation may not be a critical issue in this study as the comparison of the radiomic features was made within the selected ROIs across multiple tumors.

## 5. Conclusions

This study has demonstrated that acquired image resolution, particularly anisotropic versus isotropic, significantly affects about 30–40% of texture features, with major implications in radiomics workflows. We showed how kinetic parameter maps estimated from isotropic high-resolution DCE images provide an improved description of tumor heterogeneity through the entire volume compared with those estimated from anisotropic images. Considering the impact of acquired resolution on texture analysis, it would be prudent to acquire images at an isotropic high resolution for improved texture feature reproducibility.

**Supplementary Materials:** The following supporting information can be downloaded at: https://www.mdpi.com/article/10.3390/tomography9020058/s1, Figure S1, Percent difference and significance between features from isotropic and anisotropic (slice thickness 0.234 mm) images. (a) A heatmap depicting the percent difference in histogram and texture features between isotropic parameter maps and up-sampled anisotropic maps (slice thickness of 0.234 mm in the axial direction). (b) A heatmap showing the features that have a significant difference in radiomic features ($p < 0.00054$, depicted in red) between isotropic and anisotropic image resolutions, which are over 19.8% of all features; Figure S2, Percentage difference and significance between features from isotropic and anisotropic (slice thickness of 0.546 mm) images. (a) A heatmap depicting the percent difference in histogram and texture features between isotropic parameter maps and up-sampled anisotropic maps (slice thickness of 0.546 mm in the axial direction). (b) A heatmap showing the features that have a significant difference in radiomics features ($p < 0.00054$, depicted in red) between isotropic and anisotropic image resolutions, which are over 33.0% of all features; Figure S3, Percentage difference and significance between features from isotropic and anisotropic (slice thickness of 1.014 mm in the coronal direction) images. (a) A heatmap depicting the percent difference in histogram and texture features between isotropic parameter maps and up-sampled anisotropic maps (slice thickness of 1.014 mm in the coronal direction). (b) A heatmap showing the features that have a significant difference in radiomics features ($p < 0.00054$, depicted in red) between isotropic and anisotropic image resolutions, which are over 35.3% of all features; Figure S4, Percentage difference and significance between features from isotropic and anisotropic (slice thickness of 1.014 mm in the sagittal direction) images. (a) A heatmap depicting the percentage difference in histogram and texture features between isotropic parameter maps and up-sampled anisotropic maps (slice thickness of 1.014 mm in the sagittal direction). (b) A heatmap showing the features that have a significant difference in radiomic features ($p < 0.00054$, depicted in red) between isotropic and anisotropic image resolutions, which are over 42.1% of all features.

**Author Contributions:** Conceptualization, K.K. and S.G.K.; methodology, K.K.; software, K.K. and J.Z.; resources, S.G.K.; data curation, K.K.; writing—original draft preparation, K.K.; writing—review and editing, K.K., J.Z. and S.G.K.; visualization, K.K.; funding acquisition, S.G.K. All authors have read and agreed to the published version of the manuscript.

**Funding:** This study was supported in part by grants from the National Institute of Health (R01CA160620, R01CA219964, UG3CA22869, NIH/SIG 1S10OD018337-01, NIH/NCI 5P30CA016087, and NIH P41 EB01718).

**Institutional Review Board Statement:** The animal study protocol was approved by the Institutional Review Board (or Ethics Committee) of Weill Cornell Medicine (protocol code 2020-0028).

**Informed Consent Statement:** Not applicable.

**Data Availability Statement:** Data are available upon request.

**Conflicts of Interest:** The authors declare no conflict of interest. The funders had no role in the design of the study, in the collection, analysis, or interpretation of data, in the writing of the manuscript, or in the decision to publish the results.

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
