# Peer review of "Textural Features of Mouse Glioma Models Measured by Dynamic Contrast-Enhanced MR Images with 3D Isotropic Resolution"

_tomography, doi:10.3390/tomography9020058_

Round 1

Reviewer 1 Report

Dear Editor,

I am writing to submit my review of the manuscript entitled Textural Features of Mouse Glioma Models Measured by Dynamic Contrast-Enhanced MR Images with 3D Isotropic Resolution, which was submitted for publication in Tomography. The manuscript focuses on the impact of anisotropic resolution on the textural features of pharmacokinetic (PK) parameters in a murine glioma model using dynamic contrast-enhanced MR images. After a thorough reading and evaluation of the manuscript, I have arrived at the following conclusions: the authors have conducted a well-designed study with significant results, however, there are some areas that require clarification and revision:

  • Has a pre-processing of the images been performed before Radiomics features extraction?
  • Could authors further explain the process of ROI selection?
  • Could authors add the version of the libraries/applications used?
  • Could authors better explain the features selection strategy and the reason beneath the chosen statistical methodology?

I hope my review will be helpful to the authors and the editor in making a decision about the publication of this manuscript.

Sincerely

Reviewer 2 Report

Sample size is small. The heterogeneity in texture of the tumor cells and other parameters including response to treatment from the imaging is not well described. Isotropic imaging is helpful in order to use radiomic features. It would be very interesting to see how future studies would emerge and how imaging with the use of text analysis and radiomic features can predict malignancy and also effect of treatment.  

Reviewer 3 Report

Thank you very much for allowing me to review your manuscript. The authors present a study to test the hypothesis that the standard anisotropic acquisition of dynamic contrast-enhanced MRI alters textural radiomic features versus an isotropic approach. They designed an experiment to test that hypothesis in animal models. This is a relevant theme given the rising popularity of quantitative imaging approaches, such as radiomics, to analyze images. The results from this study could serve as a base for further human subject research, which can in turn significantly contribute to acquisition and workflow standardization in radiomics. The manuscript is well-written, fluid, and understandable by the average reader. I bring minor points that could be addressed to further improve this outstanding work.

Abstract

Nothing to add

Introduction

Lines 37-40: There have been efforts to standardize radiomics workflows some of which are even mentioned in the Discussion section. It is assumed that even anisotropic acquisitions could lead to reproducible results if the same protocol-algorithm combination is used. This could be touched upon briefly here to provide a broader view to the readers.

Materials and Methods

Lines 91-92: The manufacturer of the gadolinium contrast solution is not stated.

Lines 163-164: I would suggest rephrasing this passage to make it clearer. Which “common ROI” was used? I assume that the images were segmented only once, on the isotropic map, and then the volume was transferred to a reconstructed anisotropic acquisition. I would also advise describing in more detail the ROI drawing (i.e., program used [ITK-Snap, 3D slicer…], method (free hand, semi-automatic, linear, elliptical…)

Lines 172-173: Clarify which p-value correction strategy was adopted to reach the reported 0.0006 threshold (e.g. bonferroni correction, false discovery rate, etc).

Results

Figures 5 and 6: The fill scale is difficult to read even after zooming in. If possible, I would advise making the text elements larger.

Figure 7: The break introduced in the y axis makes the visualization less appealing and possibly misleading in a chart depicting differences in magnitude. I would suggest re-rendering the chart with a fixed and continuous y axis.

Discussion

Lines 345-346: In my understanding, manual segmentation also increases inter-observer variability.

Round 2

Reviewer 1 Report

The authors addressed all my concerns 

Author Response

We would like to thank the reviewer for catching these typos and grammatical errors. These errors have been amended and the rest of the document has been scrutinized for other errors.